# Modulation of signature cancer-related genes in oral cancer cells (Ca9-22) by anethole treatment: Insights into therapeutic potential

Meriem Hammache[1], Sara Benchekroun[1], Abdullah Alamri[2], Maroua Jalouli[3], Marwa Yousry A. Mohamed[3], Fehmi Boufahja[3], Mohamed Chahine[4], Fatiha Chandad[1], Abdelhabib Semlali [1] *

1 Faculté de Médecine Dentaire, Groupe de Recherche en Ecologie Buccale, Université Laval, Québec, QC, Canada, 2 Department of Biochemistry, College of Science, King Saud University, Riyadh, Saudi Arabia, 3 Department of Biology, College of Science, Imam Mohammad Ibn Saud Islamic University (IMSIU), Riyadh, Saudi Arabia, 4 Department of Medicine, Université Laval, Quebec City, QC, Canada

* abdelhabib.semlali@greb.ulaval.ca

**Data Availability Statement:** All relevant data are within the manuscript.

## Abstract

To explore an alternative strategy to chemotherapy to combat oral cancer, natural products and their derivates constitute one promising approach. In the last previous study, we have demonstrated the potential anti-tumor properties of anethole; an aromatic compound abundantly present in nature that serves as a major active ingredient found in plants like anise and fennel. In the current study, we aimed to investigate how this molecule inhibits oral cancer cell proliferation and induces apoptosis. This will be carried out by a transcriptomic study of its effects on the expression profile of cell cycle and apoptosis regulation genes in gingival cancer cells. cell cycle. Ca9-22 cells were treated with 10 μM of anethole (IC$_{50}$) and cell proliferation was evaluated by MTT assay. The percentage of cells in different stages of the cell cycle was measured by flow cytometry. Cytotoxicity was evaluated by LDH assay and apoptosis was investigated by Pi/Annexin V assay following 24-hour treatment. Furthermore, we employed PCR array analysis to investigate alterations in the expression levels of oncogenes and tumor suppressor genes associated with cell cycle regulation and apoptosis. Finally, Gene-gene interactions were examined using the Gene MANIA database. Our findings demonstrate that anethole significantly attenuated the proliferation of Ca9-22 cells, leading to disturbances in cell cycle progression and eliciting cellular toxicity and apoptosis. By a double normalizing with two housekeeping genes (Actin and GAPDH), we show that, treatment with 10 μM of anethole alters (more than two-fold) the expression of 13 genes involved in the control of the cell cycle (8 were up regulated and 5 were down regulated) and 7 genes involved in the regulation of apoptosis (4 were up regulated and 3 downregulated by anethole). Finally, each group of genes modulated by anethole forms a network of connections between them or with other genes. Our study suggests that anethole holds promise as a potential alternative treatment for oral cancer by its ability to modify numerous oncogenes and tumor suppressor genes implicated in the cell cycle regulation and induction of apoptosis in oral cancer cells. These findings underscore the significance of further research

**Funding:** Deanship of Scientific Research at Imam University (IMSIU) (grant number IMSIU-RP23099). The funders had no role in study design, data collection and analysis, decision to publish, or preparation of the manuscript.

**Competing interests:** The authors have declared that no competing interests exist.

**Abbreviations:** IC$_{50}$, Half maximal inhibitory concentration; LDH, Lactate Dehydrogenase; MAP kinases, A mitogen-activated protein kinase; MTT, 3-(4,5-dimethylthiazol-2-yl)-2,5-diphenyltetrazolium bromide; NF-kB, Nuclear factor kappa-light-chain-enhancer of activated B cells; OC, Oral cancer; PARP1, Poly [ADP-ribose] polymerase 1; PCR, Polymerase Chain Reaction; PPI, protein-protein interaction.

into the potential therapeutic application of anethole as an alternative drug for managing oral cancer.

## Introduction

Oral cancer (OC) is the most common tumor in the head and neck region and, when combined with pharyngeal cancer, ranks as the sixth most common cancer worldwide [1,2]. Although largely preventable, OC is attributed to multiple risk factors, including the use of tobacco in any form, excessive alcohol consumption, and betel nut and tobacco chewing [3]. The diagnostic gold standard remains a surgical biopsy of any lesion that persists beyond a two-week period without resolution [3].

Upon diagnosis, a multidisciplinary approach is employed to optimize outcomes and minimize side effects, typically involving surgical resection for staging and assessment of histopathological characteristics [4]. Radiation therapy and chemotherapy can be used as adjuvant therapy for patients with higher risks of recurrence or metastasis [4]. However, the treatment of oropharyngeal cancer carries one of the highest financial burdens of cancer treatments in the United States and is associated with significant side effects, including resistance to treatment and infertility [5,6]. Moreover, these conventional treatments may prove ineffective in achieving a cure while significantly compromising the patient's quality of life, leading to issues such as xerostomia, dysphagia, and chewing difficulties [5–7].

The limitations of current treatments and their associated impacts have driven researches to explore alternative therapies to develop more effective treatment strategies for OC [8]. In recent years, natural therapies have gained attention as complementary approaches to cancer therapy due to their potential to improve patient outcomes and reduce side effects [9]. Our research team has been investigating the potential effects of anticancer properties of trans-anethole. Anethole, formally known as 1-methoxy-4-[(1E)-prop-1-en-1-yl] benzene, is an aromatic compound found in fennel, star anise, and anise commonly used for flavoring in various industries [10]. This molecule exhibits antibacterial, antifungal, anti-inflammatory, and anti-cancer properties, yet the molecular mechanisms underlying its anti-cancer remain unclear [10,11]. Our previously published studies have demonstrated anethole's selective and dose-dependent efficacy in reducing cell proliferation and inducing cytotoxicity and apoptosis in Ca9-22 gingival cancer cells [11]. These anti-OC effects are mediated through several key cancer pathways including NF-kB, MAP kinases, caspase 3 and 9, and PARP1 [11]. Anethole also inhibits the expression of oncogenes such as cyclin D1 while upregulating cyclin-dependent kinase inhibitors (p21WAF1) [11]. Furthermore, it has been shown to induce autophagy, decrease reactive oxygen species production, and reduce intracellular glutathione activity [11]. Our previous studies also highlighted a synergistic effect between anethole and the platinum-based drug cisplatin in OC treatment. Additionally, it has been reported that anethole potentiates the effect of chemotherapy by reducing OC cell proliferation and migration by influencing multiple pathways such as NF-kB, MAP kinases, and beta-catenin [12]. These findings suggest that anethole could serve as a potential alternative or complementary therapeutic agent for OC. Despite its anti-cancer effects, its precise mechanism of action remains incompletely understood. Our earlier studies have prompted us to expand our investigation to encompass the effects of anethole on all caspases and apoptotic genes associated with both intrinsic and extrinsic pathways, as well as to explore its impact on the full array of genes regulating the cell cycle. By corroborating our initial observations with a comprehensive analysis of genes implicated in cell cycle regulation and apoptosis, we aim to further elucidate the role played by

anethole in regulating cancer cell proliferation. This endeavor underscores our commitment to advancing the understanding of anethole's therapeutic potential in combating oral cancer andcontributing to the broader landscape of cancer research.

## Material and methods

### Cell culture

The gingival epithelial squamous carcinoma cell line (Ca9-22) was procured from RIKEN BioResource Research Center (Tsukuba, Japan) and cultured in RPMI-1640 medium (Gibco; Thermo Fisher Scientific, Waltham, MA USA) supplemented with 10% fetal calf serum (Gibco; Thermo Fisher Scientific, Waltham, MA USA), 1% penicillin/streptomycin (Sigma, Oakville, ON, Canada) and an additional 1% Fungizone (Sigma, A2942). The cells were incubated at 37°C in a humidified atmosphere containing 5% $CO_2$ with the medium changed every other day until reaching 90% of cell confluence.

### Drug

Anethole, obtained from Sigma-Aldrich (Oakville, ON, Canada), was utilized at a concentration of 10 μM, consistent with the findings outlined in our previous investigations [11,13]. This concentration was determined based on its half-maximal inhibitory concentration ($IC_{50}$), as established in our prior studies.

### Cell viability by MTT assay

A total of $150 \times 10^3$ cells were seeded in 12-well plates and allowed to adhere overnight before being stimulated with 10 μM of anethole for 24 h. Following treatment, cell proliferation was measured using MTT assay as previously described in our work [11,13,14]. Upon completion of the treatment period, 1/10 v/v of MTT at 5 mg/ml was added to the cell culture and incubated at 37°C for 3 h before lysing the cells with 10 μl of isopropanol-HCl solution. The absorbance of each condition was measured in triplicate at 550 nm using a Biorad xMark microplate reader (BioRad, Mississauga, Ontario, Canada). The Ca9-22 cell viability was expressed as a percentage of cell viability following the formula: % of cell viability = [DO at 550 nm (treated cells) − DO (Blank)/DO (control cell) − DO (Blank)] × 100. The experiment was performed in triplicate 6 times.

### Cell toxicity by LDH assay

Ca9-22 cells were seeded at a density of $150 \times 10^3$ cells/well in 12-well plates and allowed to adhere overnight. Subsequently, the cells were treated with either 10 μM of anethole or left untreated for 24 h. The lactate dehydrogenase (LDH) activity was measured using an LDH kit from (Sigma-Aldrich, Oakville, ON, Canada) according to the manufacturer's instructions and protocols outlined in our previous studies [15–18]. Triton X-100 was utilized as a positive control for maximal toxicity (100%). Each experiment was conducted in triplicate (n = 6).

### Cell cycle distribution

To assess the distribution of Ca9-22 cells across different phases of the cell cycle, we employed 7-aminoactinomycin D (7-AAD), a fluorescent DNA dye from Sigma, USA, following established protocols [17].

Briefly, Ca9-22 cells were subjected to a 24-hour treatment with 10 μM of anethole. Subsequently, 1 μg/ml of 7-AAD was introduced to the cultured cells and allowed to incubate for 30 min at 37°C. The fluorescence intensities of DNA were then quantified using flow cytometry

equipment such as "LSRII" or "CantoII" from BD Biosciences. Data analysis was carried out using FACSDiva software version 6.1.3, and the experiments were conducted in triplicate, with a total of four replicates (n = 4).

## Cell apoptosis by APC Annexin V/propidium iodide assay

As previously outlined by Semlali et al. [11,13,17–19], cell apoptosis was assessed using the APC Annexin V apoptosis detection kit with propidium iodide (PI) from (Biolegend, San Diego, CA, United States). Briefly, Ca9-22 cells were seeded into T25 culture flasks and allowed to adhere overnight. Subsequently, the cells were either treated with 10 μM of anethole or left untreated for 24 h at 37°C. Following treatment, the cells were detached from the flasks using 0.05% trypsin and 0.01% EDTA then centrifuged and washed twice by PBS. The resulting pellet was resuspended in 100 μl of Annexin V binding buffer and incubated with 5 μl APC Annexin V and 5 μl of PI at room temperature for 30 min in the dark. The percentage of stained cells in various stages of apoptosis was determined by using flow cytometry analysis, employing either an "LSRII" or "CantoII" cytometer instrument from BD Biosciences. Each experiment was independently repeated four times.

## RNA extraction by TRIzol reagent

After treatment with anethole, Ca9-22 cells were suspended in 1 ml of TRIzol and stored at -80°C. Upon thawing, 200 μl of chloroform was added to 1 ml samples and incubated for 2 min to 3 min at room temperature followed by centrifugation at 1200 × g or 15 min at 4°C. The resulting transparent aqueous phase was carefully transferred to a new 1.5 ml tube and kept on ice. Subsequently, 500 μl of isopropanol 99% and 1 μl of Glycoblue were added to the sample on ice and mixed by inversion for 15 sec. The samples were then incubated at room temperature for 10 min and centrifuged at 1200 x g for another 10 min at 4°C. The supernatant was removed and 1 ml of cold ethanol 75% (-30°C) was added on ice. The sample was mixed by inversion for 15 sec and centrifuged at 12,000 × g for 5 min at 4°C. The supernatant was aspirated, and the pellet was left to dry for a maximum of 5 min under the fume hood. Following this, 10−15 μl of RNAase-free water was added and the sample was thoroughly suspended before being stored at -80°C for future use. The RNA concentration and purity were evaluated using the NanoDrop spectrophotometer (Thermo Fisher, Waltham, MA USA).

## Reverse transcriptase and gene expression using RT$^2$ Profiler PCR Array

We transcribed 2 μg of each RNA sample into cDNAs by RT$^2$ first strand kit (Qiagen, Toronto, ON, Canada) following the manufacturer's instructions. To investigate changes in gene expression related to the cell cycle, apoptosis and oncogene/tumor suppressor genes influenced by the IC$_{50}$ of anethole on gingival cancer cells, we utilized the RT$^2$ Profiler PCR Array Human Oncogenes and Tumor Suppressor Genes (Qiagen, PAHS-502Z), Human Apoptosis (Qiagen, PAHS-0127Z) and Human cell cycle (Qiagen, PAHS-020Z) from Qiagen. These arrays facilitated the assessment of the expression levels of 84 genes associated with tumorigenesis. The reaction mix for each PCR Array was prepared by combining 1350 μl of RT$^2$ SYBR Green master mix (Qiagen, Toronto, ON, Canada), 102 μl of cDNAs and 1248 μl of RNase-free water. Subsequently, 25 μl of the reaction mix was added to each 96 wells of the RT$^2$ profiler plate. The plate was securely sealed with an optical thin-wall 8-cap strips or optical adhesive film and then centrifuged for 1 min at 1000 × g at room temperature (15 C to 25 C). The CFX96 Real-Time PCR Cycler (Biorad, Mississauga, ON, Canada) was employed to analyze the expression changes. The acquired data underwent analysis using the $2^{-\Delta\Delta CT}$ method to assess differences in gene expression and fold change between untreated cells and those treated with anethole.

Our focus was on variations of at least two-fold compared to the control. CT values, fold regulation, scatter plot and heat map data were extracted and organized into a tabular format which was uploaded on an online data analysis platform at http://www.qiagen.com/geneglobe.

## Interaction protein-protein analysis

Using PCR array results as a foundation, we constructed and scrutinized a network of genes alongside their co-expressing counterparts. This analysis was facilitated by the GeneMania tool (http://www.genemania.org/), which integrates diverse data sources and databases, including protein domains, genetic interactions, co-expression patterns, predictions, and co-localization, to generate gene function predictions.GeneMania was able to forecast functionally comparable genes of hub genes and identify functionally similar genes by utilizing a range of genomics and proteomics data from a query gene. The hub genes were in the inner circle, while the predicted genes were in the outer circle.

## Western bloting

As previously described by our work {Contant, 2021 #992;Semlali, 2023 #1347;Semlali, 2021 #100, Ca9-22 cells were exposed to anethole (10 μM) for 24 h, then total proteins were extracted by lysis buffer (25 mM Tris-HCl, pH 8.0, 0.15 M NaCl, 1 mM EDTA, 10% glycerol, 0.1% SDS, 0.05% sodium deoxycholate and 1% Triton X-100). 40 μg of extracted proteins were migrated on 10% to 12% of acrylamide SDS-PAGE and transfered onto polyvinylidene difluoride membrane (PVDF) (Cytiva, Vancouver, BC, Canada). The membrane was incubated in a blocking buffer (5% non-fat milk in a solution of TBS 1x) with shaking for 1 h at room temperature followed by overnight incubation with appropriate primary antibodies: Cyclin D1 (sc-8396), p21 (sc-6246) and beta-Actin (SC-47778) were purchased from Santa Cruz Biotechnology (Santa Cruz, CA, USA). The secondary goat anti-mouse (554002) and anti-rabbit (554021) were from BD Pharmingen (Mississauga, ON, Canada). The membrane was washed 3 x10 min with TBS 1x + 0.1% Tween 20 solution. The conjugated secondary antibodies were then applied for 1 h and the membrane was washed as previously. The detection was carried out using the Clarity Western ECL Substrate (Bio-Rad, Mississauga, ON, Canada). The visualization was with VersaDoc™ MP 5000 system (Bio-Rad, Mississauga, ON, Canada). Each experiment was repeated three times.

## Statistical analysis

Each experiment was conducted a minimum of three times, and experimental values were expressed as means ± SD. The statistical significance of differences between values for control (untreated cells) and tested cells (treated with 10 μM of anethole) was determined by a percentage calculation. The statistical significance of differences between the values was assessed using a one-way ANOVA, with a p-value < 0.05 considered significant. The qPCR array analysis was performed using the $2^{-\Delta\Delta CT}$ method to evaluate differences in gene expression and fold changes between untreated cells and those treated with anethole. Normalization of gene expression was conducted through an automated process utilizing a comprehensive set of reference genes. The control samples consisted of untreated cells and the experimental group comprised cells treated with 10 μl of anethole (IC$_{50}$). For qPRCR array analysis, CT values, fold regulation, scatter plot and heat map data were extracted and organized into a tabular format which was uploaded on an online data analysis platform at http://www.qiagen.com/geneglobe. The GeneMANIA website was used to predict functionally similar genes and to establish a gene-gene interaction (GGI) network.

## Results

### Anethole exerts its anti-oral cancer effects by inhibiting cell proliferation via destabilization of cell-cycle distributions and by inducing cell cytotoxicity

To investigate the effect of $IC_{50}$ of anethole on cell viability, we performed both MTT and LDH assays. As depicted in Fig 1A and Table 1, anethole treatment reduced the viability of Ca9-22 cells by more than 40%. In parallel, the cytotoxicity of the cells was increased by 4.75-fold upon treatment with 10 μM of anethole (Fig 1B and Table 1). To elucidate the mechanisms by which anethole induces cytotoxicity, we employed flow cytometry to analyze the cell cycle distribution of Ca9-22 cells. The results, shown in Fig 1C and Table 1, indicate a significant decrease in the percentage of cells in the G0/G1phase (2-fold) by treatment with anethole. Conversely, cells in the S phase increased (5.2-fold) and those in the G2/M phase increase

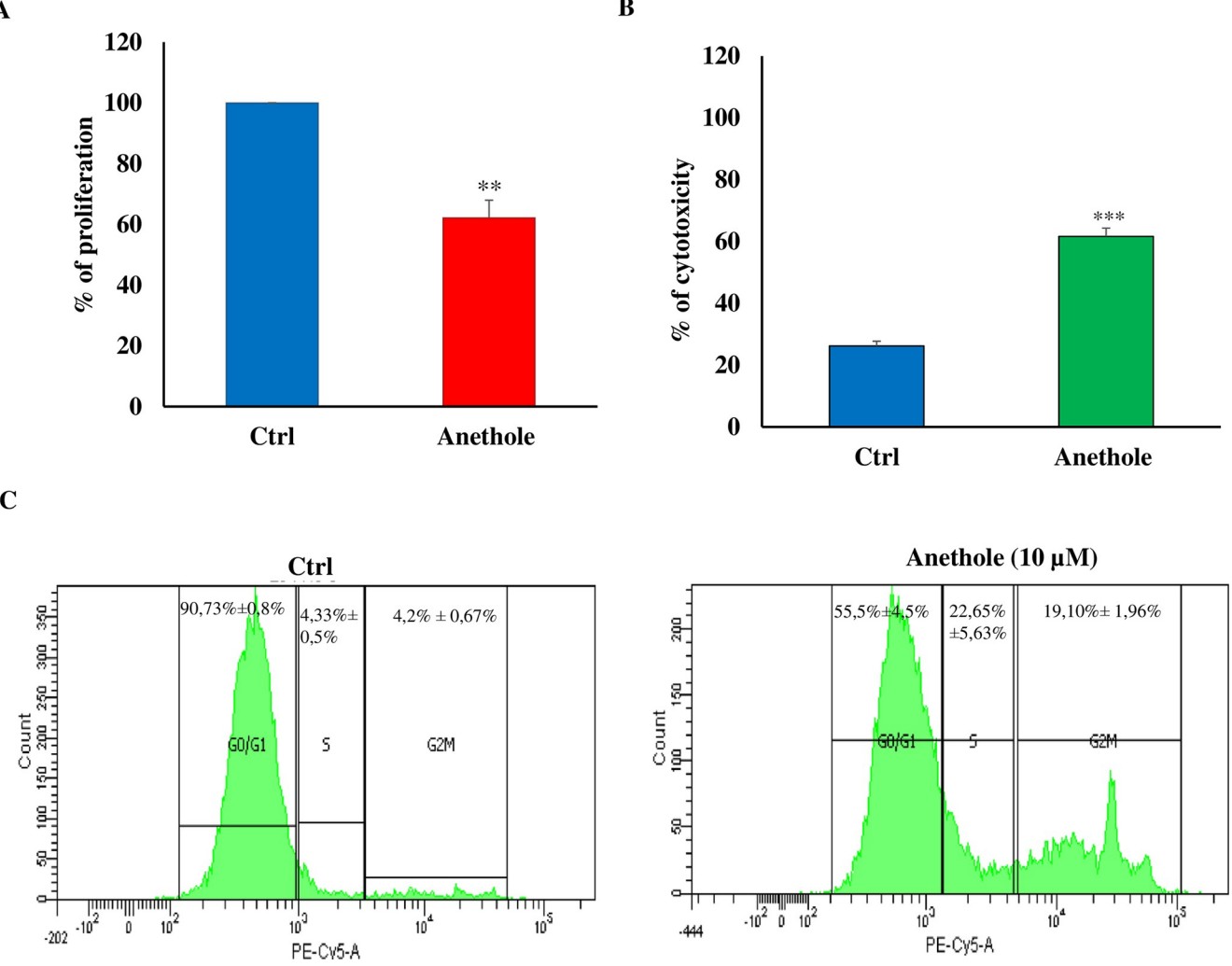

**Fig 1. Anethole treatment inhibits cell growth and induces cell cytotoxicity in Ca9-22 cells by disrupting the cell cycle.** (A) The effect of 10 μM of anethole on Ca9-22 cell proliferation by using MTT assay. The experiment was conducted six times. (B) Anethole treatment induced toxicity of Ca9-22 cell measured by LDH assay ($n = 6$). (C) Apoptosis was assessed using the PI and Annexin assay using flow cytometry (n = 4). The data was presented as the mean of percentage ± SEM. ** Correspond to p < 0.005 and *** correspond to p< 0.0005.

**Table 1. Summary data of the anethole effects on proliferation, cytotoxicity and distribution of cells in different phases of the cell cycle.** (Data in percentage).

| | % of Proliferation | % of Cytotoxicity | G0/G1 (%) | S (%) | G2/M (%) |
|---|---|---|---|---|---|
| Untreated | 100 | 26.2 ± 1.55 | 91.73 ± 0.58 | 4.33 ± 0.5 | 4.2 ± 0.67 |
| With anethole | 62.18 ± 5.77 | 81.65 ± 4.26 | 55.5 ± 4.5 | 22.65 ± 5.6 | 19.10 ± 1.96 |
| P Value | 0.003 | 0.00004 | <0.0001 | <0.0001 | <0.0001 |

(4.54-fold) in Ca9-22 cells when treated with 10 μM of anethole. These results suggest that anethole inhibition of Ca9-22 cell proliferation is due to a block in the cell cycle at the S and G2/M phases (Fig 1C).

## Anethole destabilizes cell-cycle distributions by modulating a multitude of genes involved in various phases of cell-cycle control

To identify cell cycle signature genes affected by anethole treatment in oral cancer cells, we conducted a comprehensive screening. This involved examining the expression of 84 genes associated with cell cycle regulation by using the QIAGEN cell cycle RT$^2$ Profiler PCR Arrays and by normalizing by two housekeeping gens (Actin and GAPDH). We considered a minimum of a two-fold difference as significant in our analysis when comparing Anethole-treated cells with control cells (Fig 2). Among the 84 genes investigated, known to be involved in the cell cycle, five genes were identified as being downregulated, while only eight genes showed up-regulation by at least two-fold following anethole treatment (Fig 2A). The downregulatedgenes includes: Aurora kinase A (AURKA) (-2.51-fold), cyclin G1 (CCNG1) (-4.69-fold), cyclin-dependent kinase 4 (CDK4) (-2.17-fold), retinoblastoma-binding protein 8 (RBBP8) (-2.85-fold)NBN (-2.25- fold) and stathmin 1 (STMN1) (-3.14-fold). The up regulated genes include: cyclin B1 (CCNB1)(2.3-fold), cyclin-dependent kinase inhibitor 1A(p21)(3.18-fold), cyclin-dependent kinase inhibitor 3 (CDKN3) (2.68-fold), CDC28 protein kinase regulatory subunit 2 (CKS2) (4.35-fold), cullin 2 (CUL2) (2.22-fold), MAD2 mitotic arrest deficient-like 1 (MAD2L1) (3.36-fold), cyclin H assembly factor (MNAT1) (2.43-fold) and retinoblastoma-binding protein 8 (RBBP8) (2.99-fold).(Fig 2B).

## Anethole influence on the cell cycle is explained by a protein-protein (PPI) interaction network

We utilized the GeneMANIA database and tool (available at genemania.org), and we constructed and analyzed a network of cell cycle genes and their co-expressed counterparts. This powerful tool integrates diverse data sources to predict gene functions. If two gene products were discovered to interact in a PPI study, the network displayed physical interaction; if two genes were linked and their expression levels were similar in a gene expression study, the network displayed co-expression; if there were functional relationships between genes (typically protein interactions) and if genes were expressed in the same tissue or cellular location, the network displayed co-localization. The network also reveals pathway connections when gene products participate in the same biological pathway and genetic interactions when the modification of one gene affects another. Also, shared protein domains are highlighted if gene products contain identical protein domains. Our network analysis showed seven categories that connect associated genes: shared protein domains, physical interaction, co-expression, predicted, co-localization, pathway, and genetic relationships. The examination of 13 linked genes uncovered a complex PPI network characterized by a co-expression of co-localization physical interactions, predicted interactions, pathway interaction, genetic interactions and a shared protein domain of (Fig 3). The functional significance of this network lies in its ability to

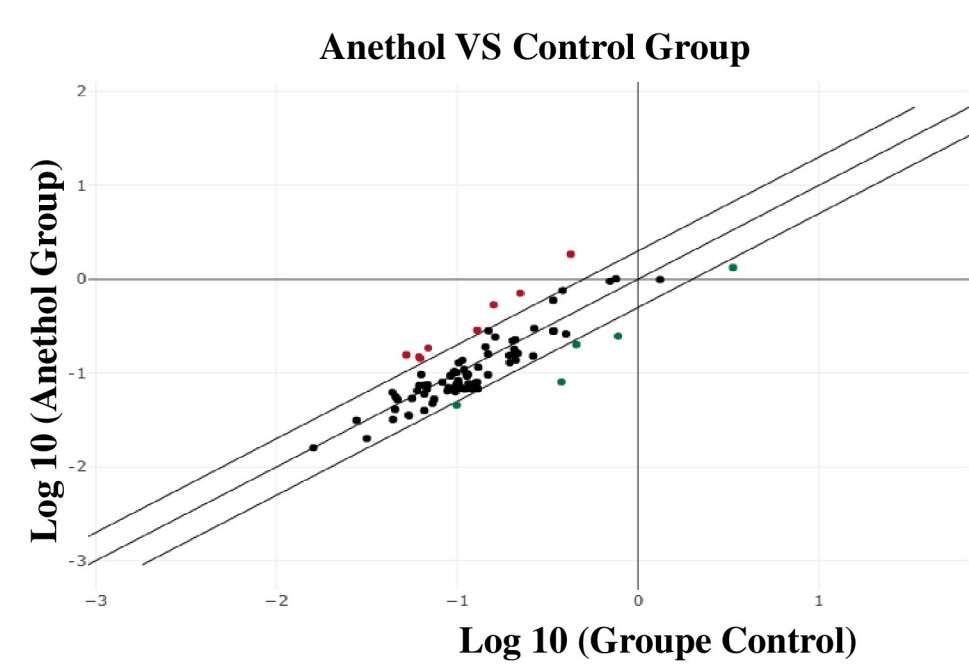

**A**

### Anethol VS Control Group

**B**

**Genes under-expressed in anethole group vs. Control group**

| Gene Symbol | Fold Regulation |
|---|---|
| AURKA | -2.51 |
| CCNG2 | -4.69 |
| CDK4 | -2.17 |
| NBN | -2.25 |
| STMN1 | -3.14 |

**Genes over-expressed in anethole group vs. Control group**

| Gene Symbol | Fold Regulation |
|---|---|
| CCNB1 | 2.30 |
| CDKN1A | 3.18 |
| CDKN3 | 2.68 |
| CKS2 | 4.35 |
| CUL2 | 2.22 |
| MAD2L1 | 3.36 |
| MNAT1 | 2.43 |
| RBBP8 | 2.99 |

**Fig 2. Anethole treatment modulated several cell cycle arrest proteins in Ca9-22.** (**A**) Cell cycle-related genes upregulated/downregulated by anethole treatment in Ca9-22 cells identified by qPCR array on 84 cell cycle markers (*n* = 3). (**B**) Overview of genes demonstrating both positive and negative modulation. Only fold regulation values exceeding 2 were taken into account (n = 3).

control critical cell cycle processes, including the G1/S phase transition, the negative regulation of the mitotic cell cycle, cell cycle checkpoints, and the negative regulation of the cell cycle transition, regulation of the G1/S transition mitotic cell cycle, regulation of the cell cycle G1/S phase transition and negative regulation of the cell cycle G1/S phase transition (Fig 3).

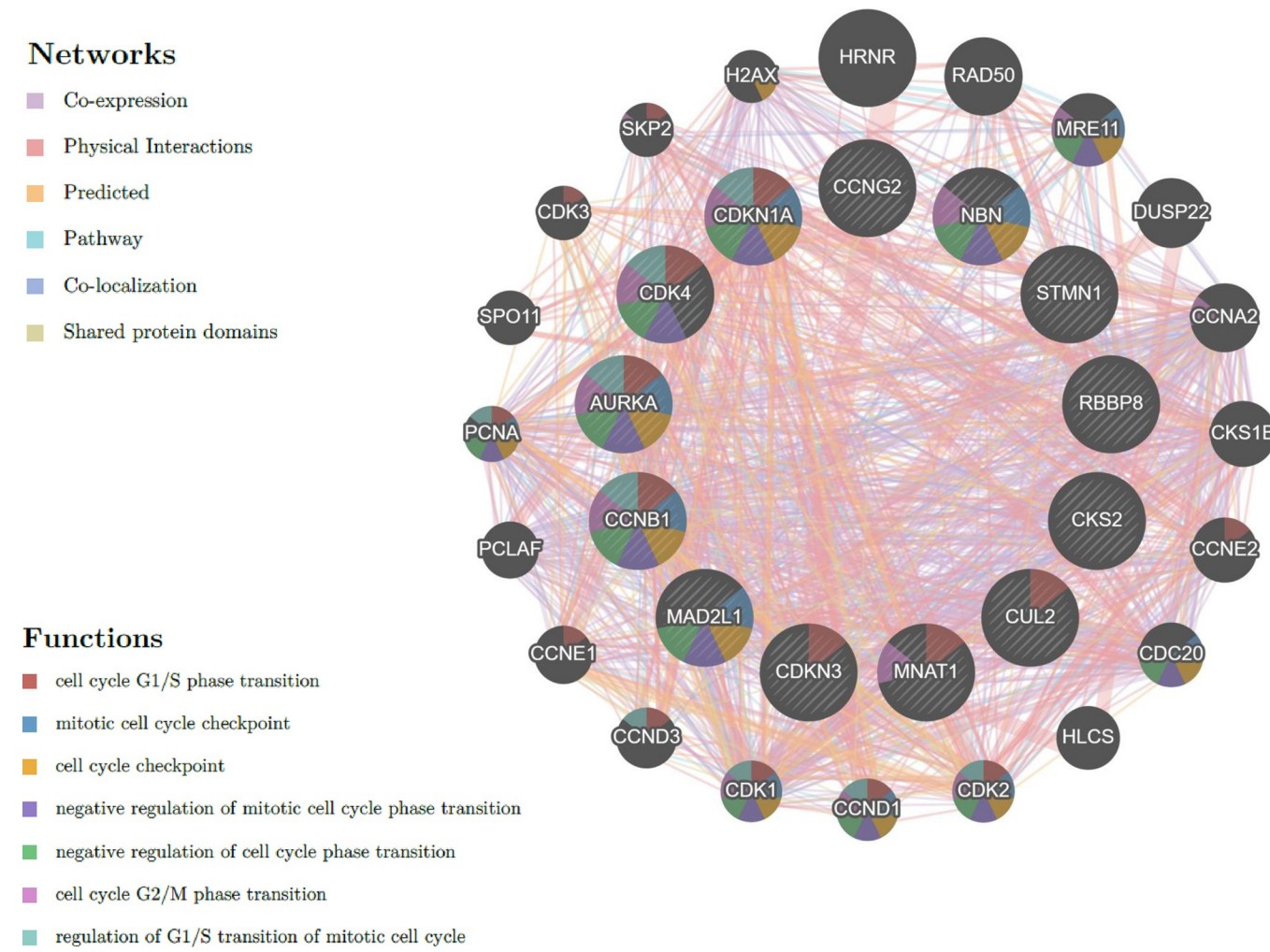

**Fig 3. Interaction networks of anethole-regulated genes involved in the cell cycle performed by the GeneMANIA database and tool available at genemania.org.**

## Anethole induces Ca9-22 oral cancer cell apoptosis

To validate the anti-cancer effect of anethole on Ca9-22 cells, we investigated the cytotoxic effects of anethole and its ability to induce apoptosis, a well-known strategy for killing cancer cells. Using flow cytometry and APV Annexin V/PI double staining, we determined the apoptotic rate of oral cancer cells treated for 24 h with 10 μM of anethole. As shown in Fig 4, the rate of live cells decreased from 70.72 ± 0.45% to 54.3 ± 1.25%. Additionally, the rate of early apoptotic cells increased from 4.33 ± 0.49% in untreated cells to 10.21 ± 1.31% in treated cells. Furthermore, the rate of late apoptotic cells increased from 7.87 ± 0.48% in control cells to 17.66 ± 1.1% post-anethole treatment. These findings highlight the potential of anethole as a promising compound for targeting oral cancer cells through apoptosis.

## Anethole induces oral cancer cell apoptosis by inhibiting 73 of 84 apoptosis genes studied by RT$^2$-PCR array and by upregulating 2 genes

To study the pro-apoptotic effects of anethole on Ca9-22 cells, we used the QIAGEN apoptosis RT$^2$ Profiler PCR Arrays. This gene screening showed that, out of 84 genes analyzed, only

**Ctrl**

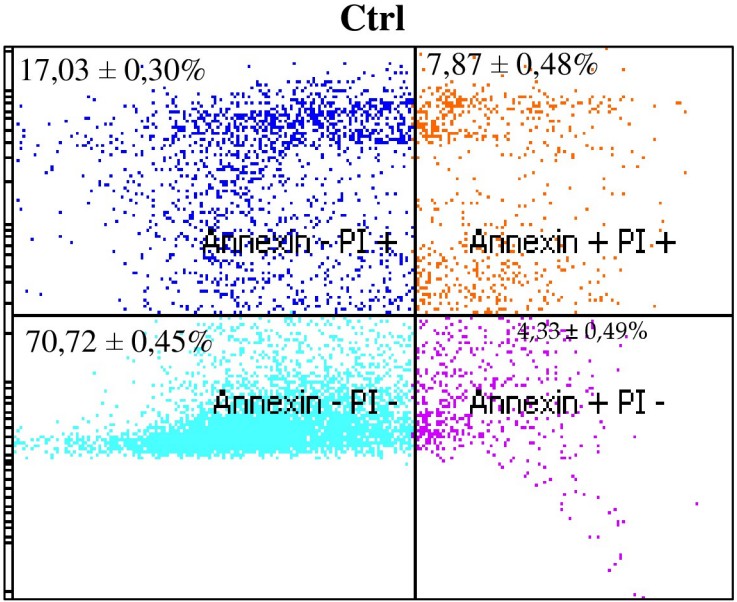

**10 µM Anethole**

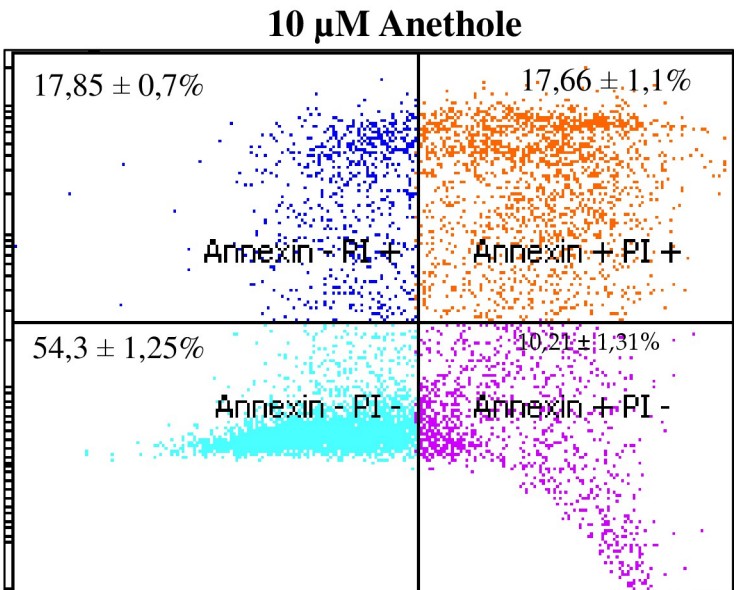

**Fig 4. Anethole treatment induces apoptosis in Ca9-22 cells, performed by flow cytometry by using APC Annexin V/PI array.** The experiment was conducted four times. The percentage of cells in different stages of apoptosis (early, late apoptosis and necrosis phase) was presented as a means of percentage ± SEM.

three were downregulated by at least a two-fold change following treatment with anethole (Fig 5A). These genes are B-cell CLL/lymphoma 10 (BCL10) (-56.30-fold), caspase 8, apoptosis-related cysteine peptidase (CASP8) (-2.42-fold), and myeloid cell leukemia sequence 1 (BCL2-related) (MCL1) (-3.67-fold). In our study of 84 genes associated with apoptosis, only four genes were found to be upregulated in the post-anethole treatment compared to the control group. These identified genes are caspase 4, apoptosis-related cysteine peptidase (CASP4)

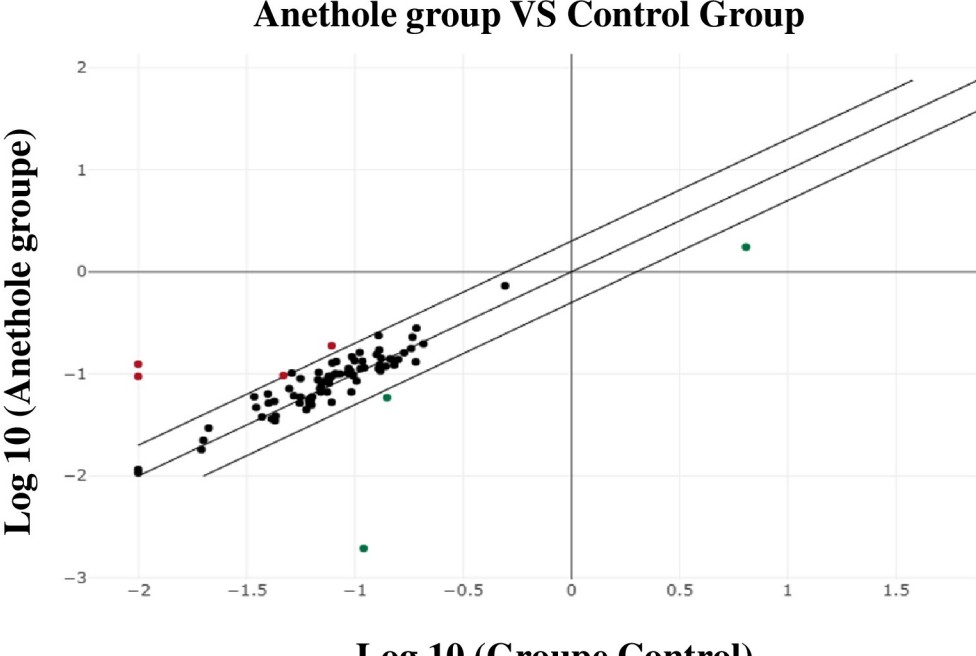

**Fig 5. Anethole treatment upregulated two genes and downregulated 73 from 84 screening apoptotic genes by qPCR array.** (**A**) Apoptosis-related genes upregulated/downregulated with anethole treatment in Ca9-22 cells identified by qPCR array on 84 cell cycle markers (n = 3). (**B**) Summary of positively and negatively modulated genes. Only fold regulation values above 2 were considered (n = 3).

2.42-fold), CD27 molecule (12.51-fold), Tumor necrosis factor receptor superfamily, member 1A (TNFRSF1A)(2.06-fold) and Tumor necrosis factor (ligand) superfamily, member 8 (TNFSF8)(9.49-fold).(Fig 5B).

### The protein-protein interaction (PPI) network is formed by anethole-modulated apoptotic genes in Ca9-22 oral cancer cells

In Fig 6, the network revealed seven categories of interactions associated with apoptotic genes regulated by anethole in Ca9-22 cells. These gene-gene interactions encompass mainly physical

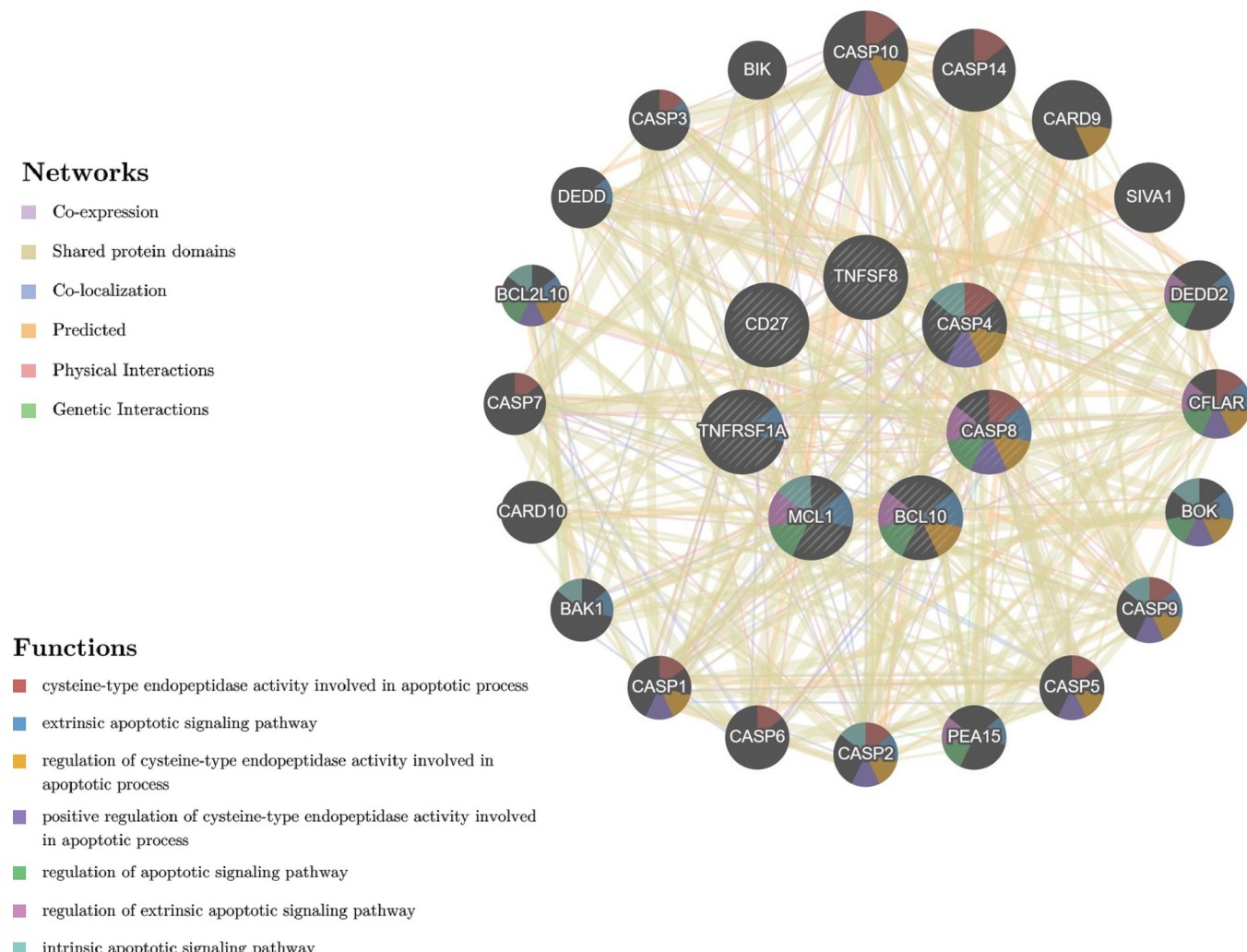

**Fig 6. Interaction networks of anethole-regulated genes involved in cell apoptosis are performed by the GeneMANIA database and tool available at genemania.org, Each type of interaction is represented by one color.**

interaction, shared protein domains, co-expression, predicted functional links, co-localization, pathway associations and genetic relationships. The complex apoptosis PPI network is involved particularly in the regulation of the apoptotic signaling pathway, negative regulation of the apoptotic signaling pathway, intrinsic apoptotic signaling pathway, extrinsic apoptotic pathway, regulation of the extrinsic apoptotic signaling pathway, positive regulation of the apoptotic process and regulation of the extrinsic apoptotic signaling pathway in absence of ligand (Fig 6). Furthermore, this network plays a functional role in controlling cell cycle dynamics, specifically the G1/S phase transition, negative regulation of mitotic cell cycle, cell cycle checkpoint, negative regulation of the mitotic cell cycle and the regulation of the G1/S transition of the mitotic cell cycle, regulation of the cell cycle G1/S phase transition and negative regulation of the cell cycle G1/S phase transition (Fig 6).

### Validation of cell cycle and apoptosis gene expression by western bloting analysis

Gene-expression analysis of 4 key genes involved in cell cycle and apoptosis was investigated to validate our qPCR Array findings. As shown in Fig 7, when Ca9-22 cells were treated with

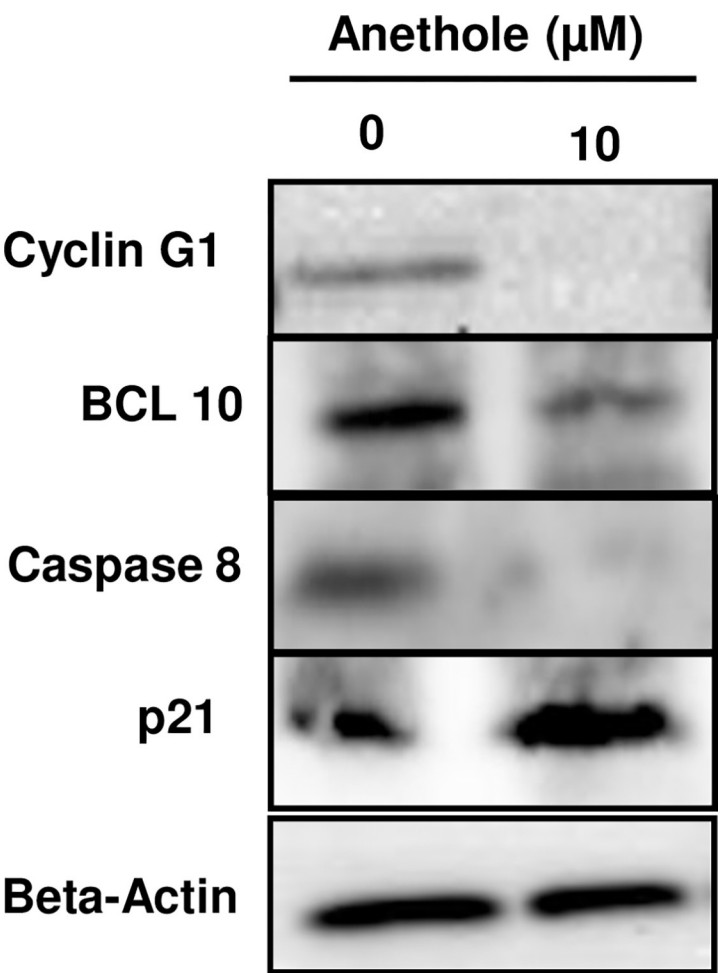

**Fig 7. Western bloting for key apoptosis and cell cycle genes modulated by anethole in Ca9-22 cells.**

10 μM of anethole, cyclin G1, BCL10 and caspase 8 were decreased by more than 2 times. However, a cyclin-dependent kinase inhibitor named p21 was significantly increased with anethole treatment. These key genes involved in cell cycle control and apoptosis regulation form an interaction network with several other genes involved in proliferation/ apoptosis.

## Discussion

The traditional approach to cancer treatment largely relies on chemotherapy, which often entails numerous adverse side effects some of which may persist chronically and lead to the development of potential resistance [6]. These challenges have motivated scientists to seek alternative treatments aimed at improving the quality of life for cancer patients and minimizing side effects. In the present study, we explored the anti-cancer mechanisms of action of anethole, a compound extracted from anise, fennel, and over 20 other plant species [11,20]. Specifically, we investigated its effects on genes that regulate the cell cycle and apoptosis. Our findings unequivocally demonstrate that a concentration of 10 μM (IC$_{50}$) of anethole inhibits cancer cell proliferation by disrupting the cell cycle, consistent with previous reports [11]. This finding supports reports indicating that anethole inhibits prostate cancer cell proliferation, clonal growth, and migration [21] by inducing the G2/M phase arrest and downregulating various cyclins such as cyclin D1, CDK-4 and c-Myc proteins while upregulating p21 and p27.

Based on the gene specificity of selected genes modulated by anethole in Ca9-22 oral cancer cells, we found that two out of the 13 analyzed genes (CDKN3 and CUL2) were involved in the G1 phase & G1/S transition. This finding is consistent with existing literature reporting that a majority of genetic alterations associated with tumorigenesis relate to the dysregulation of the G1 phase of the cell cycle [22]. Our data are in line with the literature, where targeting the increased expression of these two molecules by anethole may constitute a therapeutic strategy against oral cancer. The role of cyclin-dependent Kinase Inhibitor 3 (*CDKN3*) in promoting human tumors were reported in the literature review and pan-cancer analysis. Current research findings show that this cyclin-dependent kinase inhibitor plays a key role as a regulator of the cell cycle, as a tumor suppressor gene [23], apoptosis, tumor invasion, and metastasis [24,25]. As a cell cycle-regulating factor, CDKN3 may open up a new avenue for tumor therapy by determining its role in different tumors. In turn, CUL2 is known to preserve the balance between normal and uncontrolled proliferation. This gene seems to protect neoplastic cells from internal and external damage [26]. Increasing studies have shown that over-expression of CUL2 can prevent tumor growth and control sensitivity to cisplatin, which could constitute a new therapeutic target for cancer [27]. Moreover, our study revealed an up-regulation of the 4 genes (CCNB1,CDKN3,CKS2 and MNAT1) involved in G2phase and G2/M transition. Literature suggests that the G2/M checkpoint prevents cells from entering cellular mitosis in the presence of DNA damage, preventing the transmission of genetic defects to daughter cells [28]. Should the damage persist, the genes governing the G2/M phase may trigger apoptotic pathways [28]. Altered cell cycle control is a hallmark of tumorigenesis, rendering cell cycle regulators the prime target for therapeutic intervention. Notably, among the genes associated with the G2 phase & G2/M inhibited by anethole treatment. CCNB1. CDKN3, CKS2 and MNAT1 consist of serine/threonine kinases crucial for regulating cell cycle transition and are significantly implicated in the pathogenesis of numerous cancers [29]. Our results also demonstrate that anethole stimulation in oral cancer cells leads to the downregulation of many genes involved in the M phase. We propose that the deregulation of the M phase could lead to uncontrolled division of oral cancer cells and tumor progression [30]. The M phase is commonly targeted by chemotherapy [31]. Out of 13 genes that we analyzed, 2 genes were involved in the M phase named (AURKB and STMN1). AURKB is known to facilitate chromosome condensation and segregation and to promote cytokinesis. The overexpression of AURKB genes can lead to aneuploidy, resulting in genomic instability and a higher likelihood of tumor development. Moreover, targeting Aurora kinases has shown promising results in preclinical tumor models [32].

The cyclin family, which includes genes such as CCNG2, is known to regulate the M phase and is frequently overexpressed in various cancers [33–35]. Targeting the CDC gene family in cancer cells holds a promising potential therapeutic strategy to counteract drug resistance mechanisms and inhibit tumor growth [36–39]. This approach may represent a promising strategy for cancer therapy by inducing mitotic arrest and subsequent apoptosis in cancer cells [40]. It has been reported that the high STMN1 expression is closely associated with cancer progression and chemo-resistance in squamous cell carcinoma [41]. Concerning the genes involved in the cell cycle checkpoint and cell cycle arrest, our analysis revealed that anethole up-regulated five key genes: CCNG2, CDKN1A (p21CIP1, WAF1), CDKN3, CUL2, MAD2L1 andRBBP8. These checkpoint genes play a crucial role in regulating cell cycle progression, and mutations in these proteins are prevalent in all types of cancer, contributing to genetic instability. In preclinical studies, various checkpoint-specific inhibitors have shown promise as potential therapeutic agents. For instance, UCN-01 and its analog ICP-1 have demonstrated significant efficacy, particularly in combination with cisplatin, indicating that bypassing the intra-S phase checkpoint is indeed a pivotal step in

chemotherapy sensitization [42,43]. Five out of the 13 selected genes were involved in cell cycle regulation: AURKA, CCNB1, CDKN1A (p21CIP1, WAF1), CDKN1B (P27KIP1),. Recent discoveries in the field of cell cycle regulation and cancer have garnered significant interest [44]. The cyclin family and CDK inhibitors (p21 and p27) serve as notable examples in this domain. Understanding the molecular intricacies of protein structures, as well as protein-protein interactions within a single interaction network, could provide insights into this question and suggest strategies for cancer treatment.

The second aim of this study was to identify apoptosis signature genes modulated by anethole treatment in oral cancer cells. Targeting apoptosis is a therapeutic strategy and our previous work demonstrated that anethole can induce apoptosis by activating intrinsic pathways via caspases and PARP1 [11]. Similar results were observed in prostate cancer cells treated with anethole, where the pro-apoptotic effect was evidenced by the activation of caspase-3 and 9, DNA damage, PARP cleavage and induction of Bax/Bcl-2 protein ratio were evident [21]. Comparable results were also reported when anethole was used to treat human breast cancer cells [45]. We categorized numerous genes based on their role: those that induce apoptosis, those with anti-apoptotic effects, those involved in apoptosis regulation and those related to caspases and their regulators. Anethole was found to increase proapoptotic genes such as caspase 4 and TNFRSFs family genes including (CD27 TNFRSF1A and TNSF8) Which are considered transmembrane glycoproteins that play a key role in T and B cell costimulation. CD27 signaling is reported to plays a central immunological role, potentially usable for antitumor therapy [46]. TNFRSF1A gene encodes a member of the TNF receptor superfamily of proteins. Binding of membrane-bound TNF alpha to the membrane-bound receptor induces receptor trimerization and activation, which contributes to cell survival, apoptosis, and inflammation. Proteolytic processing of the encoded receptor results in release of the soluble form of the receptor, which can interact with free tumor necrosis factor- alpha to inhibit inflammation. TNSF8 gene is a positive regulator of apoptosis, and also has been shown to limit the proliferative potential of autoreactive CD8 effector T cells and protect the body against autoimmunity. The anti-apoptotic genes affected by anethole treatment, include BCL10 and Mcl-1 a serine-threonine kinase involved in numerous cellular pathways including cell apoptosis and angiogenesis. The BCL2 family protein known as a key regulator with pro-apoptotic and anti-apoptotic properties. TNF is a potent pro-inflammatory cytokine involved in various cancers. It was clerely reported that, evading apoptosis may promote tumor development and chemoresistance. The over-expression of prosurvival B-cell lymphoma-2 (BCL-2) family proteins like BCL10 and myeloid cell leukemia-1 (MCL1) genes constitute one of these key evasion mechanisms. MCL-1 is one of the most frequently amplified genes in cancer. Targeting MCL-1 protein is a successful strategy to induce apoptosis and overcome tumor resistance to chemotherapy and targeted therapy [47]. In addition to the genes discussed earlier, potential interactions with other cell cycle and apoptotic genes, such as BCl2, CDC34, RAD17, CDK5R1, and RAD9B among cell cycle genes, or with XRCC1, TGFβ1, MGMT, ROS1, and IGFR2 among apoptotic genes, could further enhance the complexity of our findings. The functional interactions between these genes encompass a wide array of cellular processes, including mitophagy, cell homeostasis, cell cycle progression, DNA repair, and apoptosis. Exploring these interactions could provide deeper insights into the mechanisms underlying the anticancer effects of anethole and its potential as a therapeutic agent against oral cancer." We believe that considering these potential interactions will enrich the discussion by broadening the scope of our analysis and providing a more comprehensive understanding of the multifaceted roles of anethole in cellular processes relevant to cancer development and progression.

## Conclusions

In conclusion, our treatment with anethole demonstrated significant downregulation and upregulated of several pivotal genes involved in cell cycle control and apoptosis. Targeting cell apoptosis and inhibition of proliferation presents a promising avenue for targeted cancer therapy. Furthermore, our study highlights the cytotoxic effects of anticancer drugs, primarily mediated by inducing apoptosis, which cancer cells often evade through heightened expression of anti-apoptotic proteins. Notably, our treatment impacted key anti-apoptotic genes such BCL2 family protein which has dual regulatory functions in apoptosis, and TNF, a potent pro-inflammatory cytokine involved in various cancer types. These insights offer valuable perspectives for the development of targeted cancer therapies aiming to counteract apoptosis evasion mechanisms and enhance treatment efficacy. Finally, anethole serves as a preliminary 'hit' compound for the development of a novel library of anethole analogs, with the goal to enhance selectivity and bioactivity in the treatment of oral cancer. As a future perspective, we will strategically narrow our focus to investigate the effect of anethole on the methylation status of the promoters associated with these genes, aiming to elucidate the molecular mechanisms by which anethole inhibits their expression.

## Author Contributions

**Formal analysis:** Abdullah Alamri, Marwa Yousry A. Mohamed, Fatiha Chandad.

**Investigation:** Abdelhabib Semlali.

**Methodology:** Meriem Hammache, Sara Benchekroun, Abdelhabib Semlali.

**Resources:** Fehmi Boufahja.

**Software:** Maroua Jalouli, Fehmi Boufahja, Fatiha Chandad.

**Supervision:** Abdelhabib Semlali.

**Validation:** Maroua Jalouli, Mohamed Chahine, Abdelhabib Semlali.

**Visualization:** Fehmi Boufahja.

**Writing – original draft:** Meriem Hammache.

**Writing – review & editing:** Mohamed Chahine, Abdelhabib Semlali.

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
