## [Decision Letter · Decision Letter 0]

1 Nov 2024

PONE-D-24-29904Modulation of Signature Cancer-Related Genes in Oral Cancer Cells (Ca9-22) by Anethole Treatment: Insights into Therapeutic PotentialPLOS ONE

Dear Dr. Semlali,

Thank you for submitting your manuscript to PLOS ONE. After careful consideration, we feel that it has merit but does not fully meet PLOS ONE’s publication criteria as it currently stands. Therefore, we invite you to submit a revised version of the manuscript that addresses the points raised during the review process.

**Please consider the reviewer's comments. I ask for special attention to text corrections related to English.**

We look forward to receiving your revised manuscript.

Kind regards,

Wesley Lyeverton Correia Ribeiro, Ph.D.

Academic Editor

PLOS ONE

**Journal Requirements:**

Deanship of Scientific Research at Imam University (IMSIU) (grant number IMSIU-RP23099).

The authors extend their appreciation to the Deanship of Scientific Research at Imam University (IMSIU) (grant number IMSIU-RP23099).

Deanship of Scientific Research at Imam University (IMSIU) (grant number IMSIU-RP23099).

**Additional Editor Comments:**

Please consider the reviewer's comments. I ask for special attention to text corrections related to English.

Reviewers' comments:

Reviewer's Responses to Questions

**Comments to the Author**

1. Is the manuscript technically sound, and do the data support the conclusions?

Reviewer #1: Yes

2. Has the statistical analysis been performed appropriately and rigorously? 

Reviewer #1: Yes

3. Have the authors made all data underlying the findings in their manuscript fully available?

Reviewer #1: Yes

4. Is the manuscript presented in an intelligible fashion and written in standard English?

Reviewer #1: No

5. Review Comments to the Author

**Reviewer #1: **The manuscript describes the effect of Anethole treatment on oral cancer cells (Ca9-22) using in vitro functional assays and bioinformatics. The authors examined the impact of Anethole on cell proliferation, cell cycle, and apoptosis. Moreover, PCR array analysis was employed to investigate the alterations induced by Anethole’s treatment in the overall expression of genes associated with the cell cycle, apoptosis, oncogenes, and tumor suppressor genes. Overall, the data are convincing, and those on cell proliferation and apoptosis are not new. However, the novelty of the paper resides in the PCR array and the related analysis by bioinformatics of the different genes and proteins and their interactions. From my point of view, the study lacks validation of such PCR array/bioinformatics analysis, at least for one or two key gene/protein interactions. Finally, the results section needs improvement in terms for format and style and the whole manuscript needs substantial English/grammatical checks and revision.

Here are my major comments:

- The abstract is too long and needs reshaping.

- The manuscript’s text presents many formatting issues that need to be fixed either by the authors or the journal.

- Overall, the manuscript presents many typos and unclear sentences. This needs substantial English/grammatical checks and revision.

- Everywhere, “Cell-cycle” needs to be changed to “cell cycle”.

- Line 161, the section related to RNA extraction by TRIzol reagent, must be moved one line down in a separate paragraph.

- Figure 1A/B, no need to show the structure of Anethole.

- The results section related to Figure 1 contains many % and p values. This could be summarized in a table and keep the section as text-based descriptive.

- Figure 1C is a bit confusing with different count scales (Y-axis). Visually, one would interpret that Anethole treatment increased both G0/G1 and S/G2/M population. Therefore, if both graphs had the same Y-axis scale set up to 350, G0/G1 curve should indeed be lower with Anethole treatment, BUT I AM NOT SURE for S/G2/M curves ??? It looks like S phase-curve is around 50 counts in both conditions. The authors need to clarify this and re-consider their conclusions from this experiment.

- The sections on Cell Cycle RT2 233 Profiler PCR Arrays for both cell cycle (Figure 2) and apoptosis (Figure 5) are very dense and very difficult to read with all the genes and their respective folds of change values. All this needs to be summarized in a separate table that would be related to Figures 2 and 5.

- The study lacks a functional validation of some key genes/proteins identified potentially modulated by Anethole from Cell Cycle RT2 233 Profiler PCR Arrays. Thus, since only 1 gene (the cyclin-dependent kinase inhibitor 1A) for cell cycle and 2 genes (V-raf-1 murine leukemia viral 364 oncogene homolog 1(RAF1) and retinoic acid receptor, alpha (RARA)) for apoptosis were upregulated by Anethole, I strongly ask to perform a basic validation experiment in Ca9-22 oral cancer cells where the authors may either silent these genes using siRNA or quantify their levels by western blot. Such a validation will provide an added value to the study.

- I found it surprising that none of the genes modulated by Anethole’s treatment coded for caspases, despite its demonstrated apoptotic action! Any explanation or speculation from the authors?

- In the discussion and conclusion, it’s surprising that the authors did not focus on the 3 genes (the cyclin-dependent kinase inhibitor 1A, for cell cycle, and V-raf-1 murine leukemia viral 364 oncogene homolog 1/retinoic acid receptor, for apoptosis) were upregulated by Anethole treatment. The upregulation of these genes may rationally explain the anti-cancer effects of Anethole.

6. PLOS authors have the option to publish the peer review history of their article (what does this mean?). If published, this will include your full peer review and any attached files.

Reviewer #1: No

---

## [Author Response · Author response to Decision Letter 0]

14 Nov 2024

Reviewer #1: The manuscript describes the effect of Anethole treatment on oral cancer cells (Ca9-22) using in vitro functional assays and bioinformatics. The authors examined the impact of Anethole on cell proliferation, cell cycle, and apoptosis. Moreover, PCR array analysis was employed to investigate the alterations induced by Anethole’s treatment in the overall expression of genes associated with the cell cycle, apoptosis, oncogenes, and tumor suppressor genes. Overall, the data are convincing, and those on cell proliferation and apoptosis are not new. However, the novelty of the paper resides in the PCR array and the related analysis by bioinformatics of the different genes and proteins and their interactions. From my point of view, the study lacks validation of such PCR array/bioinformatics analysis, at least for one or two key gene/protein interactions. Finally, the results section needs improvement in terms for format and style and the whole manuscript needs substantial English/grammatical checks and revision.

Here are my major comments:

- The abstract is too long and needs reshaping.

Response. The abstract was changed according to your nice suggestion (less than 350 words)

- The manuscript’s text presents many formatting issues that need to be fixed either by the authors or the journal.

Response. Thank you for this comment. The manuscript has been formatted according to the instructions of plos one

- Overall, the manuscript presents many typos and unclear sentences. This needs substantial English/grammatical checks and revision.

Response. A global revision of the manuscript was carried out by a company specializing in English editing to make it easy to read and understand. 

- Everywhere, “Cell cycle” needs to be changed to “cell cycle”.

Response. Thank you for your remark. It done

- Line 161, the section related to RNA extraction by TRIzol reagent, must be moved one line down in a separate paragraph.

Response. Sorry for this error. It done.

- Figure 1A/B, no need to show the structure of Anethole.

Response. I agree. The structure was removed from figure 1 as suggested.

- The results section related to Figure 1 contains many % and p values. This could be summarized in a table and keep the section as text-based descriptive.

Response. As suggested, Table 1 was added in section 1 of results.

- Figure 1C is a bit confusing with different count scales (Y-axis). Visually, one would interpret that Anethole treatment increased both G0/G1 and S/G2/M population. Therefore, if both graphs had the same Y-axis scale set up to 350, G0/G1 curve should indeed be lower with Anethole treatment, BUT I AM NOT SURE for S/G2/M curves ??? It looks like S phase-curve is around 50 counts in both conditions. The authors need to clarify this and re-consider their conclusions from this experiment.

Response. Yes the figure is correct and we have added the percentages of each population in each figure to avoid confusion

- The sections on Cell cycle RT2 233 Profiler PCR Arrays for both cell cycle (Figure 2) and apoptosis (Figure 5) are very dense and very difficult to read with all the genes and their respective folds of change values. All this needs to be summarized in a separate table that would be related to Figures 2 and 5.

Response. Thank you for your nice suggestion. The figure 2 and fig 5 were changed and we have introduced a new table to simplify the understanding of data

- The study lacks afunctional validation of some key genes/proteins identified potentially modulated by Anethole from Cell cycle RT2 233 Profiler PCR Arrays. Thus, since only 1 gene (the cyclin-dependent kinase inhibitor 1A) for cell cycle and 2 genes (V-raf-1 murine leukemia viral 364 oncogene homolog 1(RAF1) and retinoic acid receptor, alpha (RARA)) for apoptosis were upregulated by Anethole, I strongly ask to perform a basic validation experiment in Ca9-22 oral cancer cells where the authors may either silent these genes using siRNA or quantify their levels by western blot. Such a validation will provide an added value to the study.

Response. We have already anticipated this question and we have added a figure 7 shown the western blotting analysis for the expression of 4 key genes involved in cell cycle and apoptosis. In next study we will investigate the role of miRNA on expression of theses genes and also to stydy their profile of methylation. Thank you for your nice suggestion. 

- I found it surprising that none of the genes modulated by Anethole’s treatment coded for caspases, despite its demonstrated apoptotic action! Any explanation or speculation from the authors?

Response. I agree your suggestion, In figure 7 we have added the expression of Caspase 8 down regulated by anethole

- In the discussion and conclusion, it’s surprising that the authors did not focus on the 3 genes (the cyclin-dependent kinase inhibitor 1A, for cell cycle, and V-raf-1 murine leukemia viral 364 oncogene homolog 1/retinoic acid receptor, for apoptosis) were upregulated by Anethole treatment. The upregulation of these genes may rationally explain the anti-cancer effects of Anethole.

Response. The conclusion section was changed as suggested by reviewers . Also results section and figures according to new analysis by normalizing by two housekeeping genes

---

## [Decision Letter · Decision Letter 1]

21 Nov 2024

Modulation of Signature Cancer-Related Genes in Oral Cancer Cells (Ca9-22) by Anethole Treatment: Insights into Therapeutic Potential

PONE-D-24-29904R1

Dear Dr. Semlali,

We’re pleased to inform you that your manuscript has been judged scientifically suitable for publication and will be formally accepted for publication once it meets all outstanding technical requirements.

Kind regards,

Wesley Lyeverton Correia Ribeiro, Ph.D.

Academic Editor

PLOS ONE

Additional Editor Comments (optional):

Reviewers' comments:

Reviewer's Responses to Questions

**Comments to the Author**

1. If the authors have adequately addressed your comments raised in a previous round of review and you feel that this manuscript is now acceptable for publication, you may indicate that here to bypass the “Comments to the Author” section, enter your conflict of interest statement in the “Confidential to Editor” section, and submit your "Accept" recommendation.

Reviewer #1: All comments have been addressed

2. Is the manuscript technically sound, and do the data support the conclusions?

Reviewer #1: Yes

3. Has the statistical analysis been performed appropriately and rigorously? 

Reviewer #1: Yes

4. Have the authors made all data underlying the findings in their manuscript fully available?

Reviewer #1: Yes

5. Is the manuscript presented in an intelligible fashion and written in standard English?

Reviewer #1: Yes

6. Review Comments to the Author

Reviewer #1: All the comments were addressed by the authors. The revised manuscript is now suitable for publication in PLOS ONE.

7. PLOS authors have the option to publish the peer review history of their article (what does this mean?). If published, this will include your full peer review and any attached files.

Reviewer #1: No

---

## [Editor Report · Acceptance letter]

8 Dec 2024

PONE-D-24-29904R1 

PLOS ONE

Dear Dr. Semlali, 

I'm pleased to inform you that your manuscript has been deemed suitable for publication in PLOS ONE. Congratulations! Your manuscript is now being handed over to our production team.

Kind regards, 

on behalf of

Dr. Wesley Lyeverton Correia Ribeiro 

Academic Editor

PLOS ONE